# Interventions to improve racial and ethnic equity in critical care: A scoping review

Shirley Ge[1,2¤], Hope Lappen[3], Luz Mercado[2], Kaylee Lamarche[2], Theodore J. Iwashyna[4], Catherine L. Hough[5], Virginia W. Chang[2,6], Adolfo Cuevas[2], Thomas S. Valley[7,8,9,10], Mari Armstrong-Hough[2,11]*

1 Albert Einstein College of Medicine, Montefiore Medical Center, Bronx, New York, United States of America, 2 Department of Social and Behavioral Sciences, School of Global Public Health, New York University, New York, New York, United States of America, 3 Division of Libraries, New York University, New York, New York, United States of America, 4 Departments of Medicine and Health Policy and Management, Johns Hopkins University, Baltimore, Maryland, United States of America, 5 Division of Pulmonary and Critical Care Medicine, Department of Medicine, Oregon Health and Science University School of Medicine, Portland, Oregon, United States of America, 6 Department of Population Health, Grossman School of Medicine, New York University, New York, New York, United States of America, 7 Institute for Healthcare Policy and Innovation, University of Michigan, Ann Arbor, Michigan, United States of America, 8 Division of Pulmonary and Critical Care Medicine, Department of Internal Medicine, University of Michigan, Ann Arbor, Michigan, United States of America, 9 VA Center for Clinical Management Research, Ann Arbor, Michigan, United States of America, 10 Center for Bioethics and Social Sciences in Medicine, University of Michigan, Ann Arbor, Michigan, United States of America, 11 Department of Epidemiology, School of Global Public Health, New York University, New York, New York, United States of America

¤ Current address: Stamford Hospital, Columbia University Vagelos College of Physicians & Surgeons, Stamford, Connecticut, United States of America.
* mah842@nyu.edu

## Abstract

### Background

Racial and ethnic disparities in the delivery and outcomes of critical care are well documented. However, interventions to mitigate these disparities are less well understood. We sought to review the current state of evidence for interventions to promote equity in critical care processes and patient outcomes.

### Methods

Four bibliographic databases (MEDLINE/PubMed, Web of Science Core Collection, CINAHL, and Embase) and a list of core journals, conference abstracts, and clinical trial registries were queried with a pre-specified search strategy. We analyzed the content of interventions by categorizing each as single- or multi-component, extracting each intervention component during review, and grouping intervention components according to strategy to identify common approaches.

### Results

The search strategy yielded 11,509 studies. Seven-thousand seventeen duplicate studies were removed, leaving 4,491 studies for title and abstract screening. After

**Data availability statement:** All relevant data are within the manuscript and its Supporting Information files.

**Funding:** The author(s) received no specific funding for this work.

**Competing interests:** The authors have declared that no competing interests exist.

screening, 93 studies were included for full-text review. After full-text review by two independent reviewers, eleven studies met eligibility criteria. We identified ten distinct intervention components under five broad categories: education, communication, standardization, restructuring, and outreach. Most examined effectiveness using pre-post or other non-randomized designs.

## Conclusions

Despite widespread recognition of disparities in critical care outcomes, few interventions have been evaluated to address disparities in the ICU. Many studies did not describe the rationale or targeted disparity mechanism for their intervention design. There is a need for randomized, controlled evaluations of interventions that target demonstrated mechanisms for disparities to promote equity in critical care.

## Introduction

Disparities in the delivery and outcomes of critical care are well documented [1–5]. Black patients are less likely to receive timely or guideline-concordant antibiotics [6], Hispanic patients are more likely to receive deep sedation, and both Black and Hispanic patients are less likely to receive timely tracheostomy [7–9]. Black and Hispanic patients also have higher rates of sepsis and mortality from acute respiratory distress syndrome compared to White patients [10,11]. Furthermore, Black patients have higher incidence of venous thromboembolism and noncardiogenic respiratory failure [12–15].

Racial and ethnic disparities in outcomes of pediatric critical care are also well recognized. In the neonatal intensive care unit (NICU) setting, non-White infants have a two-fold greater risk of mortality related to intraventricular hemorrhage (IVH) and are less likely to survive necrotizing enterocolitis (NEC) [16,17]. Non-White infants are also less likely to be discharged on breastmilk feeding, which is associated with lower incidence of IVH, NEC, neurodevelopmental delays, late-onset sepsis, and chronic lung disease [18–23].

As the coronavirus disease 2019 pandemic disproportionately sickened Black, Hispanic, and Native American populations, awareness of inequities in critical care and interest in interventions to combat disparities increased [24,25]. However, despite longstanding recognition of disparities in outcomes of critical care and increasing urgency to address them, few interventions to reduce disparities have been routinely deployed in the critical care setting [2,3]. We therefore aimed to review existing and ongoing studies of interventions that target racial or ethnic disparities in the delivery or outcomes of ICU care. The objective of this scoping review is to consolidate the available knowledge regarding interventions and quality improvement initiatives to reduce racial and ethnic disparities in critical care.

## Methods

### Literature search

This scoping review addressed the question, "What is the current state of evidence for interventions to reduce racial and ethnic disparities in critical care processes and

patient outcomes?" To answer this question, we developed a review protocol based on a framework by Visintini, Evidence Synthesis Coordinator at the Maritime SPOR SUPPORT Unit [26]. A research librarian (HL) was consulted in study design and results analysis. The Preferred Reporting Items for Systematic Reviews and Meta-Analyses extension for Scoping Reviews (PRISMA-ScR) standard was followed in reporting procedures and results [27,28] (S1 Checklist).

We included past and ongoing studies, trials, and initiatives with an interventional component that aimed to address racial/ethnic disparities in medical, surgical, and pediatric intensive care settings in the United States. We limited the scope of our review to the United States to enhance comparability of studies. Studies were excluded if they aimed only to detect the presence of a racial or ethnic disparity, lacked an intervention, were conducted outside the U.S., were not published in English, or did not take place in an intensive care setting or examine critical illnesses.

An iterative search methodology was employed to optimize the comprehensiveness of our literature search. First, four bibliographic databases (MEDLINE/PubMed, Web of Science Core Collection, CINAHL, and Embase) were queried with pre-specified search strategies. Journals, conference abstracts, and clinical trial registries were hand-searched using the same criteria. All sources were searched from inception until December 31, 2023 (S1 Database).

Databases were searched using an iterative approach that began with a focused search strategy which included terms on improvement, reduction, and intervention to quickly identify relevant articles (S1 Database). Diseases and devices common to the ICU setting (acute respiratory distress syndrome (ARDS), respiratory failure, extracorporeal membrane oxygenation (ECMO), and sepsis) were incorporated into search strategy to ensure any studies related to critical care medicine were captured. This initial, focused strategy is shown to demonstrate the overall structure of the search strategies. The search strategies for each database and iteration are reported in S1 Database:

(race OR racial OR ethnic*)

AND

(disparity OR disparities OR inequality OR inequalities OR inequity OR inequities)

AND

(icu OR critical care OR intensive care OR intensive care units OR critical illness OR ARDS OR respiratory failure OR ECMO OR sepsis)

AND

(quality improvement OR improvement OR improve OR reduction OR reduce OR intervention OR intervene).

Though "inequity" has a distinct meaning from "inequality" or "disparity", health equity depends on addressing inequalities and disparities [29,30]; thus, the search terms "inequity" and "inequities" were used to capture all possible articles relevant disparity research. Following the first focused search, the databases were queried again using broader search strategies that removed terms specifying an interventional component and also integrated subject headings and additional ICU-related keywords to ensure the search was comprehensive.

After surveying the databases, sources that report ongoing studies were hand-searched. These included ClinicalTrials.gov, NIH RePORTER, MedRXiv, and the Veterans Affairs (VA) Health Services Research and Development citations database. The VA database reports both past and ongoing studies. Journals and conference abstracts from Society of Critical Care Medicine and the American Thoracic Society were also hand-searched for reports of past or ongoing studies. The resulting literature was imported into Covidence systematic review software for deduplication, screening, and extraction [31]. The studies were screened by title and abstract by one reviewer (SG). The remaining literature was full-text reviewed by two team members (SG, MAH) independently. Conflicts within the full-text screening regarding study exclusion or the specific reason for exclusion were discussed between the two reviewers and a third reviewer (HL) and resolved by

consensus or a tie-breaking vote by the third reviewer. The reasons for exclusion were recorded per PRISMA-ScR guidelines. The two reviewers (SG, MAH) then each independently extracted the included studies and compared the extractions to reach a final consensus.

## Analysis

Both reviewers extracted detailed characterizations of the interventions evaluated in the included studies. Because there is heterogeneity in how race and ethnicity were characterized and measured by investigators, reviewers also recorded how each study defined its target disparity and how demographic data were collected. Next, reviewers catalogued each intervention component and distinguished between complex interventions that bundled together multiple intervention components and simple interventions that consisted of a single intervention component [32]. Reviewers then discussed each intervention component and organized them into a taxonomy of intervention types and elements. The taxonomy was developed by grouping similar intervention components under a broad category and then further categorizing the components into more granular, specific classifications. By creating a taxonomy, we could compare the different interventional approaches among the included studies and identify patterns in current and previous interventions aimed at disparities.

## Results

### Literature search

For the first focused search, bibliographic database queries and hand-searches yielded 1,406 studies (Fig 1, S1 Database). The broad search of the databases resulted in 5,021 articles, and the second broad search with subject headings and additional terms resulted in 5,072 articles. Seven thousand seventeen duplicates were removed, leaving 4,491 studies for title and abstract screening of which 4,398 studies were deemed irrelevant. One report was merged because it was the study protocol for another study already added for screening [33,34]. After full-text review of the remaining 93 studies by two independent reviewers, 11 studies met eligibility criteria.

During full-text review, studies were excluded if they did not include a prospective assessment of an intervention (S1 Text). Eighty-two studies were excluded during full-text review, of which 34 studies (41%) were disqualified for ineligible study design. These studies lacked an intervention and either focused on investigating the mechanisms for disparities or discussed potential ways to address disparities without carrying out an evaluation. Twenty-one studies (26%) were excluded for "Formative research (no intervention)", meaning they sought to detect or explore a racial/ethnic disparity but did not pilot an intervention to resolve the disparity. Two studies (2%) were omitted for "Wrong intervention" because the study intervention did not address a racial or ethnic disparity. In one study, investigators retrospectively assessed if a previously implemented protocol to reduce sepsis mortality had equitably benefited patients of racial or ethnic minority background, as opposed to designing and implementing an intervention to reduce a previously observed disparity [35]. The second excluded study examined how using a screening tool for social determinants of health in the NICU impacted patients and families, but did not directly address associated health disparities by race and ethnicity [36]. The 11 studies that remained after full-text review each had an interventional component to address a racial or ethnic disparity in care processes or outcomes in an intensive care setting in the United States.

### Intervention targets and results

Table 1 summarizes the 11 included studies. Nine were single-center studies. Two were multicenter studies: the Massachusetts Human Milk QI Collaborative by Parker et al., a statewide QI initiative that involved all 10 level III NICUs in Massachusetts [37], and the Mayo Clinic Care Network (MCCN) Guiding Coalition by Linnander et al., a coalition-based leadership intervention across eight U.S. health systems and their surrounding communities [38]. Six studies (55%) were

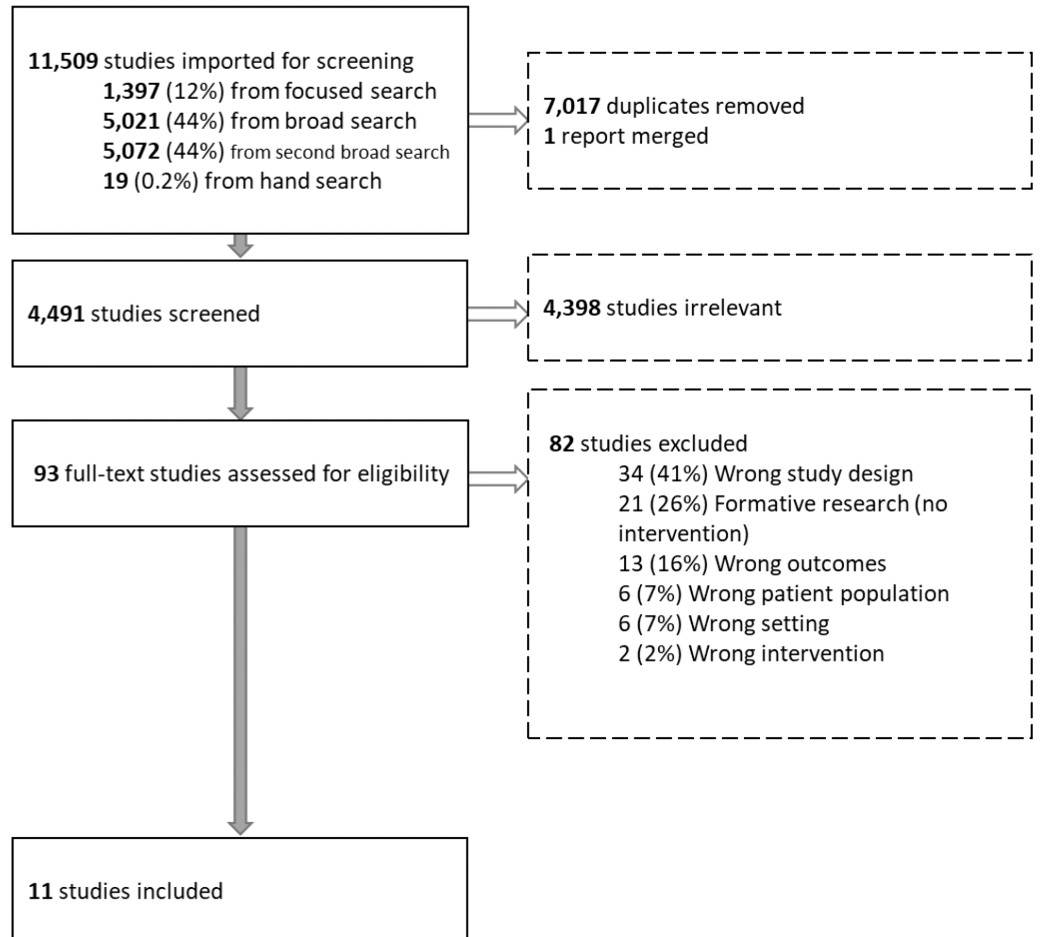

| Exclusion Criteria | Operational Definition |
|---|---|
| Wrong study design | Lacked an intervention and either focused on investigating the mechanisms for disparities or discussed ways to potentially address disparities in critical care without an evaluation. |
| Formative research | Did not pilot or evaluate an intervention to resolve an identified racial/ethnic disparity. |
| Wrong outcomes | Did not focus on reducing disparities. |
| Wrong patient population | Did not focus on critically ill patients. |
| Wrong setting | Did not take place in an intensive care setting or examine critical illness. |
| Wrong intervention | Intervention did not address a racial or ethnic disparity. |

**Fig 1. PRISMA-ScR Flowchart.**

ongoing clinical trials [34,38–42]. Two studies (18%) employed interventions that continued to be implemented past the study period [37,43].

More than half of studies (six studies, 55%) evaluated interventions for pediatric or newborn populations, with four (36%) studies targeting disparities in mother's-own-milk feedings in the NICU [34,37,42,43]. The two remaining pediatric studies targeted ICU mortality and parent-infant verbal interactions in the NICU, respectively [44,45]. In the five studies of interventions to reduce disparities in adults with critical illness (45%), four studies focused on improving communication

**Table 1. Population Summary.**

| Study Title & First Author | Outcome Measure(s) | Publication Type & Date | Type of Study & Funding Sources | Study Population | Race Definition | Pre-Intervention Period? | Intervention(s) Tested & If There Was a Pre-Intervention Period |
|---|---|---|---|---|---|---|---|
| **Impact of a Communication Intervention Around Goals of Care on Racial Disparities in End-of-Life Care in the Intensive Care Unit** Anne Mosenthal | Cardiopulmonary resuscitation, do-not-resuscitate, withdrawal of life support, and length of stay | Conference abstract, 2010 | Retrospective cohort study, no funding sources | Black and non-Black MICU patients, presumably adults. | No mention on how race data were obtained. | No | 1. Interdisciplinary rounds with prognosis evaluation 2. Facilitated family meetings around goals of care for those patients identified with poor prognosis. |
| **Pediatric intensive care unit mortality among Latino children before and after a multilevel health care delivery intervention** Kanwaljeet J. S. Anand | Odds of mortality at PICU Discharge | Peer-reviewed journal article, 2015 | Observational study, no funding sources | Pediatric patients (<= 18 yo) discharged from the PICU at a tertiary care metropolitan hospital. | Race data was obtained from medical records. Only White, Black, and Latino children were included. | Yes | 1. Education of health care professionals on cultural competent care 2. Recruitment of more bilingual staff 3. Availability of 24-hour interpreter services in the emergency department and PICU 4. Translation of consent forms and educational materials for patients and families 5. Culturally sensitive end-of-life care discussions, with participation of palliative care services 6. Outreach efforts and contacts with the Latino community to remove barriers to health care access 7. Help from the city government and local health department and ment for preventive services. |

*(Continued)*

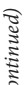

Table 1. (Continued)

| Study Title & First Author | Outcome Measure(s) | Publication Type & Date | Type of Study & Funding Sources | Study Population | Race Definition | Pre-Intervention Period? | Intervention(s) Tested & If There Was a Pre-Intervention Period |
|---|---|---|---|---|---|---|---|
| Implementation of a Reading Program in the Intensive Care Nursery Decreases Socio-economic and Racial Disparities in Parent-infant Verbal Interactions: A Pilot Model for Early Language Intervention Among High Risk Preterm Infants<br><br>Laura H. Rubinos | Parent self-reports of reading to their infant | Conference abstract, 2018 | Qualitative research, funding sources not mentioned | Parents of infants less than 33 weeks gestational age at an academic level III NICU. | No mention on how race was defined or how data were collected. | No | 1. Ongoing access to developmentally appropriate books through creation of an in-unit family lending library 2. Incorporation of parental educational resources and anticipatory guidance on importance of early literacy 3. Completion of a 10-question anonymous parent survey on admission, discharge, and during the developmental outpatient follow up visit. |
| Addressing Disparities in Mother's Milk for VLBW Infants Through Statewide Quality Improvement†<br><br>Margaret G. Parker | Any and exclusive mother's milk in the 24 hours before the initial disposition | Peer-reviewed journal article, 2019 | Quality improvement study, funding from W.K. Kellogg Foundation | Very low birth-weight infants or infants<30 weeks' gestation. | No mention on how race was defined. | Yes | Interventions on a variety of targets (parental education, early initiation of milk expression, inadequate continuation of milk, transition to direct breastfeeding, 'other', and racial ethnic disparities) were implemented. The use of these interventions varied between NICUs. Statewide interventions on disparities included 1. Created 4 multicultural education handouts for families in 9 languages 2. Created 10 multicultural education videos for families, 5 in English and 5 in Spanish 3. Qualitative interviews with non-Hispanic black and Hispanic mothers |

*(Continued)*

Table 1. (Continued)

| Study Title & First Author | Outcome Measure(s) | Publication Type & Date | Type of Study & Funding Sources | Study Population | Race Definition | Pre-Intervention Period? | Intervention(s) Tested & If There Was a Pre-Intervention Period |
|---|---|---|---|---|---|---|---|
| **Targeting Bias to Reduce Disparities in End of Life Care (BRiDgE)\*** Elizabeth Chuang | Ratio of clinician to patient speaking time | Clinical trial registration, 2021 | Randomized controlled trial, funding from Montefiore Medical Center | Physicians who treat patients with serious illness. | N/A, intervention implemented on providers | Yes | Training session to improve communication skills and reduce the effect of racial bias on clinician communication behavior based on transformational learning theory. |
| **Improving racial disparities in unmet palliative care needs among intensive care unit family members with a needs-targeted app intervention: The ICUconnect randomized clinical trial\*** Christopher E. Cox | Change in unmet palliative care needs measured by a scoring instrument between baseline and 3 days post-randomization | Peer-reviewed journal article (protocol manuscript), 2021 | Randomized controlled trial, funding from National Institute on Minority Health and Health Disparities | Non-Hispanic Black or non-Hispanic White adults in the ICU for >= 48 hours. | Race data were obtained from medical records. | No | ICUconnect mobile app which provides a digital infrastructure to guide the needs-focused interaction between family members and clinicians, promotes family engagement by providing a question coaching feature, and links families to reliable information on ICU practices and therapies |

*(Continued)*

| Study Title & First Author | Outcome Measure(s) | Publication Type & Date | Type of Study & Funding Sources | Study Population | Race Definition | Pre-Intervention Period? | Intervention(s) Tested & If There Was a Pre-Intervention Period |
|---|---|---|---|---|---|---|---|
| "Liquid Gold" Lactation Bundle and Breastfeeding Rates in Racially Diverse Mothers of Extremely Low-Birth-Weight Infants†<br><br>Maria Obaid | Exclusive mother's own milk (MOM) diet at discharge | Peer-reviewed journal article, 2021 | Quality improvement study,<br><br>funding from the Kellogg Foundation and the Heckscher Foundation | Black, White, and Hispanic extremely low birthweight infants. | Race data obtained from medical records. | Yes | 1. Staff education consisting of a 30-minute online training module with pre- and post-tests<br>2. Standardized daily skin-to-skin care protocol<br>3. COC within 24 hours of birth<br>4. Assistance with expression of breast milk within 6 hours postpartum<br>5. HM as the first enteral feed (MOM or PDHM)<br>6. Provision of exclusive HM diet supported by PDHM and HM fortifier (Prolact+H2 MF)<br>7. Daily rounds to assess lactation progress with a multidisciplinary team<br>8. Access to a certified LC in the L&D unit and for postpartum counseling<br>9. Provision of food trays for mothers when they visited their babies in the NICU<br>10. Elimination of all formula promotional materials<br>11. Assistance with obtaining a double electric breast pump, as mandated by the Affordable Care Act and covered by private and Medicaid insurance plans<br>Post-2016:<br>12. Implementation of HM cream (Prolact+CR)—an HM-derived caloric fortifier intended to supply more lipids and achieve adequate growth<br>13. Addition of a Spanish-speaking LC and increase in the full-time equivalent LC support to 2.0. |

*(Continued)*

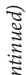

Table 1. (Continued)

| Study Title & First Author | Outcome Measure(s) | Publication Type & Date | Type of Study & Funding Sources | Study Population | Race Definition | Pre-Intervention Period? | Intervention(s) Tested & If There Was a Pre-Intervention Period |
|---|---|---|---|---|---|---|---|
| **Improving Lactation Success in Black Mothers of Critically Ill Infants***<br><br>Leslie Parker | Number of participants acceptance of the intervention | Clinical trial registration, 2021 | Randomized controlled trial,<br><br>funding from University of Florida | Black mothers of newborn infants admitted to the NICU | Race was self-reported. | No | A hands-free, wearable breast pump with an associated app that tracks pumping frequency and breast milk production to increase lactation success |
| **Reducing Disparity in Receipt of Mother's Own Milk in Very Low Birth Weight Infants (ReDiMOM)***<br><br>Aloka L. Patel | Receipt of MOM at NICU Discharge | Clinical trial registration, 2021 | Randomized controlled trial,<br><br>funding from National Institute on Minority Health and Health Disparities | Mothers and hospitalized infants in NICU. Only Black, Hispanic, and White patients were included. | Race was self-reported. | No | 1. Hospital-grade electric smart breast pump for home use at no charge to the mother while the infant is in the NICU and the mother continues to pump<br><br>2. Free pickup of expressed MOM from home to transport to NICU 2–3 times per week during weekdays as needed<br><br>3. Receives payment for opportunity costs of pumping and handling milk at $18.50/day for each day that the mother pumps during her infant's NICU stay |
| **Mitigating Racial Disparities in Shared Decision Making in the Intensive Care Unit***<br><br>Deepshikha Ashana | Adoption rates of best practices for shared decision making with families of patients with acute respiratory failure | Clinical trial registration, 2022 | Non-randomized experimental study,<br><br>funding from National Heart, Lung and Blood Institute | Critically ill patients, non-Hispanic Black and White family members, and ICU physicians | Race was self-reported. | No | Tip sheets containing best practices for shared decision making with diverse families |

*(Continued)*

**Table 1.** (Continued)

| Study Title & First Author | Outcome Measure(s) | Publication Type & Date | Type of Study & Funding Sources | Study Population | Race Definition | Pre-Intervention Period? | Intervention(s) Tested & If There Was a Pre-Intervention Period |
|---|---|---|---|---|---|---|---|
| Mitigating structural racism to reduce inequities in sepsis outcomes: a mixed methods, longitudinal intervention study*<br><br>Erika L. Linnander | Early identification of sepsis as measured by time to antibiotic, in-hospital mortality, and 30-day hospital readmission | Peer-reviewed journal article (protocol manuscript), 2022 | Non-randomized experimental study,<br><br>funding from National Institute of General Medical Sciences to Yale University | 8 U.S. health systems and surrounding communities | Socially assigned race obtained from medical records/EHR. | Yes | A coalition-based leadership intervention that will include<br><br>three components:<br><br>1. a series of five semiannual virtual, cross-site forums<br><br>2. a series of four one-day workshops onsite at each hospital<br><br>3. a web-based platform to allow sites to share experiences and to serve as a repository for program resources. |

*Current ongoing study

†Interventions that continued to be implemented past the study period

between patients and providers for end-of-life care and shared-decision making [39–41,46]. Only one study targeted a clinical diagnosis, specifically sepsis [38]. Additionally, four out of five of the adult studies were ongoing at the time of analysis and had not yet reported any results [38–41].

Most studies focused on increasing salutary health behaviors for all patients in practices that have been shown to have race-based differences [34,37,39,43,44,46]. For example, Obaid et al.'s primary objective in their "Liquid Gold" intervention was to increase mother's own milk feedings in the NICU for all patients and their secondary aim was to evaluate the impact of the intervention on disparities between African American, White, and Hispanic mother-infant pairs [43]. In contrast, only Anand et al.'s study sought to resolve disparities in patient outcomes, in particular, reducing mortality in a Latino pediatric population [45].

Five studies (45%) had a pre-intervention period in which they assessed sources or mechanisms of disparities in the target outcome to inform the design of the intervention (Table 1) [37,38,41,43,45]. For instance, in Linnander et al.'s ongoing initiative to address inequities in sepsis outcomes, coalitions comprised of hospital leadership, patient advocacy representatives, and community organizations are participating in forums and workshops to identify which inequity processes are present in their health system and ways to address them. Coalitions will then implement the strategies and continually evaluate and adjust their solutions over a 2.5-year study period [38].

Linnander et al.'s study was also unique because it targeted organizational change to reduce racial and ethnic disparities [38]. While most interventions were not based on behavioral theories, Chuang's clinical trial cited transformational learning theory as the basis of its intervention [41].

Lastly, studies varied in how they defined race and collected race data on participants (Table 1). Patient race was generally obtained from medical records or self-reported by patients; however, some studies did not specify a data collection method.

## Taxonomy analysis

Our analysis of the intervention elements that appeared across studies yielded a taxonomy of 10 intervention components under five broad categories (Tables 2 and 3). The categories were *Education* (e.g., education of healthcare providers and/or education of patients and families), *Communication* (e.g., facilitated communication between providers and patients and increasing number of bilingual staff/and or interpreter services), *Standardization* (e.g., creating an automatic clinical decision support tool within the electronic medical record and standardizing a protocol for a specific healthcare process), *Restructuring* (e.g., financial and/or material resources provided to patient and families and restructuring patients' environment), and *Outreach* (e.g., outreach efforts to the community and help from government services). Seven studies (64%) included *Education* elements, six (55%) *Communication*, three (27%) *Standardization*, five (45%) *Resourcing*, and only two (18%) *Outreach* (Table 3).

The characterization of intervention components captured the multifactorial approaches investigators used to address disparities (Tables 2 and 3). Six studies (55%) employed complex "bundled" interventions, which combined two or more intervention components to achieve a desired outcome (Table 2) [37,40,43–46]. The pediatric studies combined more intervention components than adult studies. The four pediatric studies with complex interventions (36%) had a range of three to seven components in their bundles [37,43–45]. In contrast, the two adult studies (18%) evaluating complex interventions each included two intervention components [40,46]. The adult studies also did not include any *Standardization*, *Resourcing*, or *Outreach* elements. Across studies, the most commonly deployed intervention component was education of healthcare professionals (six studies, 55%) and facilitated communication between providers and patients and families (six studies, 55%), followed by financial and/or material resources provided to patient and families (four studies, 36%).

## Discussion

In this scoping review, we described and analyzed existing interventions to reduce racial or ethnic disparities in critical care settings. Despite decades of evidence identifying racial and ethnic disparities in ICU care [1], our findings suggest

Table 2. Complex Intervention Components.

| | | PEDIATRIC STUDIES | | | | | | ADULT STUDIES | | | | |
|---|---|---|---|---|---|---|---|---|---|---|---|---|
| | | DPI Anand et al. | RPI Rubinos et al. | LGL Obaid et al. | ADM Parker et al. | ILS Parker et al. | RDR Patel et al. | CIE Mosenthal et al. | MSR Linnander et al. | ICU Cox et al. | TBR Chuang et al. | MRD Ashana et al. |
| Education | Education of health-care professionals | X | | X | X | | | | X | | X | X |
| | Education of patient & families | X | X | | X | | | | | | | |
| Communication | Facilitated communication between providers & patients & families | X | | X | X | | | X | | X | | X |
| | Increasing number of bilingual staff and/or interpreter services | X | | X | | | | | | | | |
| Standardization | Automatic clinical decision support tool within the electronic medical record | | | | X | | | | | | | |
| | Standardized and delivered a protocol for specific healthcare element | | X | X | X | | | | | | | |
| Resourcing | Financial and/or material resources provided to patient & families | | | X | X | X | X | | | | | |
| | Restructuring patients' environment | | X | X | | | | | | | | |
| Outreach | Outreach efforts to the community | X | | | | | | | | | | |
| | Help from government services | X | | X | | | | | | | | |

**LEGEND**

| | |
|---|---|
| DPI | Decreased Pediatric ICU Mortality Intervention. Kanwaljeet J. S. Anand et al. |
| RPI | Reading Program in the Intensive Care Nursery. Laura H. Rubinos et al. |
| LGL | Liquid Gold Lactation Bundle in ELBW Infants. Maria Obaid et al. |
| ADM | Addressing Disparities in Mother's Milk Through Statewide QI. Margaret G. Parker et al. |
| ILS | Improving Lactation Success in Black Mothers of Critically Ill Infants. Leslie Parker et al. |
| RDR | Reducing Disparity in Receipt of Mother's Own Milk (ReDiMOM). Aloka L. Patel et al. |
| CIE | Communication Intervention in End of Life Care. Anne Mosenthal et al. |
| MSR | Mitigating structural racism to reduce inequities in sepsis outcomes. Erika L. Linnander et al. |
| ICU | The ICUconnect randomized clinical trial. Christopher E. Cox et al. |
| TBR | Targeting Bias to Reduce Disparities in End-of-Life Care (BRiDgE). Elizabeth Chuang et al. |
| MRD | Mitigating Racial Disparities in Shared Decision Making in the Intensive Care Unit. Deepshikha Ashana et al. |

**Table 3. Intervention Components.**

| Intervention Category | Number of unique studies (n = 11) |
|---|---|
| **Education** | **7 (64%)** |
| Education of healthcare professionals | 6 (55%) |
| Education of patient and families | 3 (27%) |
| **Communication** | **6 (55%)** |
| Facilitated communication between providers and patients and families | 7 (64%) |
| Increasing number of bilingual staff and/or interpreter services | 2 (18%) |
| **Standardization** | **3 (27%)** |
| Automatic clinical decision support tool within the electronic medical record | 1 (9%) |
| Standardized and delivered a protocol for specific healthcare element | 3 (27%) |
| **Resourcing** | **5 (45%)** |
| Financial and/or material resources provided to patient and families | 4 (36%) |
| Restructuring patients' environment | 2 (18%) |
| **Outreach** | **2 (18%)** |
| Outreach efforts to the community | 1 (9%) |
| Help from government services | 2 (18%) |

that very few interventions to address these recognized disparities in the ICU have been systematically evaluated. After searching databases of peer-reviewed articles, conference abstracts, and ongoing study registries, our review yielded only 11 intervention studies. Previous systematic reviews suggest that comparatively more attention has been paid to identifying disparities in critical care than to evaluating interventions to reduce disparities. To illustrate, Soto et al. and McGowan et al. found 38 and 25 studies, respectively, that identified disparities in adult ICU care [1,2]. Sigurdson et al. found 41 studies that identified disparities in neonatal ICU care [3]. Soto et al. noted also that no studies had yet aimed to reduce disparities at the time of publishing in 2013, and similarly, the other two groups did not describe intervention studies [1–3].

Kilbourne et al. organized health disparities research into three phases: detecting disparities, understanding the underlying mechanism of a disparity, and designing interventions to reduce and eliminate disparities [47]. The relative scarcity of intervention-oriented studies found in our study compared to the large number of observational studies found in prior reviews demonstrates that ICU disparities research has largely remained in the detection phase. Though research focused on detecting disparities has demonstrated significant differences in ICU care and outcomes by race and ethnicity, interventions designed to reduce or eliminate disparities remain troublingly rare. Even using a broad definition of intervention, we found only 11 intervention studies aiming to reduce racial or ethnic disparities in outcomes or delivery of critical care, with eight studies being from the last five years. These include reports for six ongoing randomized, controlled trials (RCTs) of interventions to reduce disparities in critical care settings [34,38–42].

One possible explanation for this stall is that researchers have yet to closely study mechanisms underlying disparities in critical care outcomes. Many studies did not describe the interventions they evaluated as designed around an empirically grounded theory or formative mechanistic research linking a particular barrier (e.g., clinician knowledge) to a specific patient outcome (e.g., increased rates of infants receiving mother's own milk). Though some investigators designed interventions in response to observations during a pre-intervention period, assessments from the pre-study period and rationale for the associated intervention component were not consistently reported. Most studies clearly identified disparities and then proposed a bundle of intervention components that might reduce disparities through multiple pathways. For

example, in Anand et al.'s study to reduce mortality of Latino PICU patients, investigators conducted a pre-intervention assessment and subsequently employed a bundle intervention, but did not elaborate the reasoning behind each intervention component [45]. Consequently, it is difficult to distinguish effective intervention components from ineffective components. Moreover, interventions designed using behavior change frameworks are more effective than interventions designed without frameworks [48,49]. Only Chuang's study cited a behavioral theory for the basis of the intervention [41]. Future research should review the state of knowledge regarding mechanisms driving critical care disparities to develop theoretically and empirically grounded interventions.

Our scoping review did not seek to assess the quality of study designs because we aimed to survey current knowledge on reducing disparities in the ICU and, as our review demonstrates, research to reduce disparities is at an early state [28]. There is a breadth of study types among the included literature, including RCTs, pre-post studies, and QI initiatives (Table 1). Thus, it would not have been reasonable to evaluate study quality. It is, however, notable that most studies relied on pre-post designs rather than RCT designs. There are many barriers to carrying out randomized, controlled evaluations of interventions to reduce disparities. Most interestingly, many interventions to reduce disparities are organizational and must be delivered at the level of the ICU or the hospital rather than at the level of the patient. Stepped-wedge and other cluster-randomized, controlled trials are appropriate for evaluating such interventions, but they are resource-intensive. Nonetheless, more than half of the studies included in our analysis are currently ongoing trials, which suggests that controlled trials of interventions to reduce inequalities in critical care are growing rapidly.

We found that many of the interventions included in this review focused on standardizing delivery of care with the aim of reducing disparities in patient outcomes. Structured care processes can reduce bias in care and clinical decision-making. However, the instruments and measures used to deliver standardized care may themselves need to be re-evaluated to avoid re-inscribing structural bias. For example, the incorporation of race into the equation of estimated glomerular filtration rate (eGFR) is debated because the formula assigns higher eGFR values to Black patients without substantial empirical data. Inflating the kidney function of Black patients affects medication doses and restricts patients from receiving timely nephrology referrals, participating in clinical trials, and obtaining life-saving kidney transplants [50]. Due to these implications, many health institutions now mandate calculation of eGFR using a formula without a race parameter. In addition, studies have also shown that pulse oximeters produce biased measures of oxygen saturation for patients with darker skin tone [51]. Care processes that make use of biased instruments may worsen quality of care and endanger minority patients even when clinicians themselves do not exercise bias in clinical decision-making.

We also found that the adult interventions to reduce disparities in adult critical care used a relatively narrow range of strategies. Interventions for adult critical care focused on educating providers regarding culturally competent care and facilitating communication with patients. In contrast, the studies seeking to reduce disparities in pediatric critical care employed more intervention types, including strategies that extended beyond the ICU such as *Outreach* and *Resourcing* to address structural sources of disparities. This difference may be attributed to the fact that parents or other adult guardians are typically surrogate decision makers for pediatric patients, and so pediatric intensivists are more confronted by their patients' social contexts. The more varied approach of the pediatric studies highlights possible new approaches for interventions in adult critical care.

Finally, methodology for collecting race and ethnicity data was not consistently reported across studies. The American Medical Association guidance for reporting race and ethnicity recommends that authors report in their Methods an explanation describing both how race and ethnicity data were collected and why they were collected in this way [52]. Future studies should report the method and rationale for identifying the race and ethnicity of participants.

Our review possessed several strengths. We systematically searched a wide variety of databases to identify both completed and ongoing research, as well as research that may have been carried out but never reported in peer-reviewed venues. Furthermore, our approach to extraction of intervention targets and components for each study enabled us to

identify the most common strategies across a broad range of interventions to reduce disparities. Finally, our methods relied on multiple independent reviewers to improve rigor and reliability.

Our scoping review also had some limitations. First, we were unable to identify ongoing studies not registered in Clinicaltrials.gov or NIH RePORTER. Privately funded studies or quality improvement initiatives may be missing from our analysis if they were not reported in a peer-reviewed journal. Second, due to the small number of studies and our objective of providing a comprehensive review of all critical care interventions, we included interventions targeting age-diverse populations, from neonates to adults. The NICU is a unique care setting and the barriers and facilitators to equal care delivery and outcomes in this setting may differ substantially from those in pediatric or adult critical care. In addition, our review only included studies conducted in the United States, so interventions employed outside of the U.S. or published in non-English may have been overlooked. We did not include non-U.S studies because the U.S. has a distinct landscape of racial and ethnic disparities in healthcare, shaped by its history, policies, and sociocultural dynamics. Studies outside the U.S. would likely address disparities within different sociopolitical and healthcare contexts, making direct comparisons challenging [25,53–55].

## Conclusion

Despite the widespread recognition of racial and ethnic disparities in critical care, there is little evidence of rigorous efforts to address them through interventions. This is the first study to comprehensively examine interventions addressing racial and ethnic disparities in critical care processes and patient outcomes. Of the few interventions that have been evaluated, many did not describe the rationale or targeted disparity mechanism for their intervention design. Interventions should be developed based on theories that address recognized mechanisms of disparities. There is a need for randomized, controlled evaluations of theory-informed interventions to reduce racial and ethnic disparities in critical care.

## Supporting information

**S1 Checklist. Preferred Reporting Items for Systematic reviews and Meta-Analyses extension for Scoping Reviews (PRISMA-ScR) Checklist.**
(DOCX)

**S1 Database. Database Search Strategies.**
(DOCX)

**S1 Text. Studies Excluded in Full-Text Review.**
(DOCX)

## Author contributions

**Conceptualization:** Shirley Ge, Theodore J. Iwashyna, Catherine L. Hough, Thomas S. Valley, Mari Armstrong-Hough.

**Data curation:** Shirley Ge, Hope Lappen, Mari Armstrong-Hough.

**Formal analysis:** Shirley Ge, Hope Lappen, Mari Armstrong-Hough.

**Investigation:** Shirley Ge, Luz Mercado, Mari Armstrong-Hough.

**Methodology:** Shirley Ge, Hope Lappen, Thomas S. Valley, Mari Armstrong-Hough.

**Project administration:** Shirley Ge.

**Supervision:** Mari Armstrong-Hough.

**Visualization:** Kaylee Lamarche.

**Writing – original draft:** Shirley Ge, Hope Lappen, Luz Mercado, Mari Armstrong-Hough.

**Writing – review & editing:** Shirley Ge, Hope Lappen, Luz Mercado, Kaylee Lamarche, Theodore J. Iwashyna, Catherine L. Hough, Virginia W. Chang, Adolfo Cuevas, Thomas S. Valley, Mari Armstrong-Hough.

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
