## [Decision Letter · Decision Letter 0]

21 Jan 2025

Dear Dr. Ge,

Thank you for submitting your manuscript to PLOS ONE. After careful consideration, we feel that it has merit but does not fully meet PLOS ONE’s publication criteria as it currently stands. Therefore, we invite you to submit a revised version of the manuscript that addresses the points raised during the review process.

We look forward to receiving your revised manuscript.

Kind regards,

De-Chih Lee, Ph.D.

Academic Editor

PLOS ONE

Journal Requirements:

Additional Editor Comments :

Please make minor revisions based on the comments of the two reviewers.

Reviewers' comments:

Reviewer's Responses to Questions

**Comments to the Author**

1. Is the manuscript technically sound, and do the data support the conclusions?

Reviewer #1: Yes

Reviewer #2: Yes

2. Has the statistical analysis been performed appropriately and rigorously?

Reviewer #1: Yes

Reviewer #2: N/A

3. Have the authors made all data underlying the findings in their manuscript fully available?

Reviewer #1: Yes

Reviewer #2: No

4. Is the manuscript presented in an intelligible fashion and written in standard English?

Reviewer #1: Yes

Reviewer #2: Yes

Reviewer #1: There are two limitations of this study that have not been discussed:

1) "We limited the scope of our review to the United States to enhance comparability of studies."

2) "Our scoping review did not seek to assess the quality of study designs because, as our review demonstrates, research to reduce disparities is at an early stage."

These two inclusion criteria could have limited external validity and increased the risk of bias. These two aspects should be highlighted in the discussion, and limitations regarding interpretation should be addressed.

Even though inequity has been included as a search term, it has not been discussed, and references to use this perspective are missing.

Reviewer #2: Thank you for conducting this incredibly valuable work, and for the opportunity to review it for the team. My expertise lies in knowledge synthesis methodology generally and search methodology specifically, so that is where I will be focusing my feedback.

Data availability: this paper has good advice for what kind of data could/should be shared for knowledge syntheses. While you provide a lot, I think there's some "behind the curtain" stuff that could be shared, eg. templates, rough extraction data, list of full-text references that didn't meet inclusion, etc.

Page MJ, Nguyen PY, Hamilton DG, Haddaway NR, Kanukula R, Moher D, et al. Data and code availability statements in systematic reviews of interventions were often missing or inaccurate: a content analysis. J Clin Epidemiol [Internet]. 2022;147:1–10. Available from: https://doi.org/10.1016/j.jclinepi.2022.03.003

Methods:

- the PRISMA you cite is for systematic reviews, please follow/cite the Scoping Review extension

- technically PRISMA is a reporting guideline only, not conduct. For conduct see JBI and Arksey & O'Malley.

- on page 4 you mention journals, conference abstracts and clinical trial registries, change subjects, and then report on the handsearching again - consider tightening this section up for clarity, and potentially making use of tables in the supplementary files (e.g. source searched, dates, keywords/method used to search, etc)

- does it make sense to report two database search approaches if ultimately you conducted the broader search anyway? I feel like this is eating up potential word count for results/discussion

- (pg.6) PROSPERO is a systematic review protocol registry, does the team mean PRISMA?

- please review PRISMA-S and report search strategies accordingly. They should be provided in full as run in the supplemental files. For example from how the searches are presented, it's hard to know if subject headings were searched in addition to keywords, what fields were searched, etc.

- I'm a little concerned that the search may be overly simplistic (see for example the racial/ethnic search filters available here: https://sites.google.com/a/york.ac.uk/issg-search-filters-resource/home/population-specific)

Results:

- I really like the way the team chose to break down results. It's very clear and the groupings are helpful for getting a bird's eye view.

- your tables are really clear and easy to read

Discussion:

- the Kilbourne phases is a helpful breakdown

- I think it's important here to emphasize that these are only American findings, and there may be different levels of development/interrogation of mechanism of action in other jurisdictions that are similar to the US (e.g. Canada, UK, Australia), or just globally, since every country has their own racial minorities. It would be helpful to contextualize your findings with the rest of the world, or if that's not feasible, frequently emphasize that these findings only relate to the US and not all of critical care literature.

- quality appraisal in scoping reviews are not common because they're intended to answer different questions. It might be simpler to just cite JBI here rather than getting into breadth of study designs.

- I think your decision to limit to US-only is also a limitation, since NICUs/ICUs are pretty structured settings (it's not like we're comparing public health initiatives or something really nebulous) and could have had a lot to offer and provided richer context of where the literature is right now.

Final thoughts: This was a really well written and thoughtful paper, and provides some very rich feedback for researchers in this field moving forward. I'm a bit concerned about the robustness of the search, especially if subject headings weren't used, as that is standard practice for knowledge synthesis searching. This would be easy enough to fix, though it would take this from a minor revisions to a major one since it would require additional screening, etc.

**Do you want your identity to be public for this peer review?** For information about this choice, including consent withdrawal, please see our Privacy Policy

Reviewer #1: **Yes: ** Airton Tetelbom Stein

Reviewer #2: No

---

## [Author Response · Author response to Decision Letter 1]

7 Mar 2025

Reviewer #1: There are two limitations of this study that have not been discussed:

1) "We limited the scope of our review to the United States to enhance comparability of studies."

2) "Our scoping review did not seek to assess the quality of study designs because, as our review demonstrates, research to reduce disparities is at an early stage."

These two inclusion criteria could have limited external validity and increased the risk of bias. These two aspects should be highlighted in the discussion, and limitations regarding interpretation should be addressed.

We have added both these limitations to the Discussion. We now also elaborate on the decision to limit our literature search to the United States because the U.S. has a distinct landscape of racial and ethnic disparities in healthcare shaped by its history, policies, and sociocultural dynamics (Lines 369-375).

Even though inequity has been included as a search term, it has not been discussed, and references to use this perspective are missing.

In Methods, we now describe how “inequity” is distinct from “inequality” or “disparity” and cite references for these distinctions. We then justify inclusion of “inequity” and “inequities” in our search strategy to capture all articles relevant to disparity research (Lines 124-126).

Reviewer #2: Thank you for conducting this incredibly valuable work, and for the opportunity to review it for the team. My expertise lies in knowledge synthesis methodology generally and search methodology specifically, so that is where I will be focusing my feedback.

Data availability: this paper has good advice for what kind of data could/should be shared for knowledge syntheses. While you provide a lot, I think there's some "behind the curtain" stuff that could be shared, eg. templates, rough extraction data, list of full-text references that didn't meet inclusion, etc.

Page MJ, Nguyen PY, Hamilton DG, Haddaway NR, Kanukula R, Moher D, et al. Data and code availability statements in systematic reviews of interventions were often missing or inaccurate: a content analysis. J Clin Epidemiol [Internet]. 2022;147:1–10. Available from: https://doi.org/10.1016/j.jclinepi.2022.03.003.

We address this by adding new supplementary materials, including the PRISMA-S checklist and a list of the papers excluded during full-text review (see Supplementary Materials 1 and 3).

Methods:

- the PRISMA you cite is for systematic reviews, please follow/cite the Scoping Review extension

- technically PRISMA is a reporting guideline only, not conduct. For conduct see JBI and Arksey & O'Malley.

Thank you; we have clarified the citation of the PRISMA reporting guideline used. We have also clarified the language in the Methods section to clarify that we used the PRISMA-S guideline for reporting our procedures and results, not for guiding our procedures (Lines 91-94).

- on page 4 you mention journals, conference abstracts and clinical trial registries, change subjects, and then report on the handsearching again - consider tightening this section up for clarity, and potentially making use of tables in the supplementary files (e.g. source searched, dates, keywords/method used to search, etc).

We created Supplementary Material 2 which details the search strategy formatted to query each database, how many articles each database yielded, and further details on the search terms and yields of the hand search portion of our review. We also clarified the language in this paragraph (Lines 106-107).

- does it make sense to report two database search approaches if ultimately you conducted the broader search anyway? I feel like this is eating up potential word count for results/discussion.

We included both search strategies to highlight how we sought to specifically search for interventions targeting health disparities in critical care. As our study was a scoping review and we aimed to survey all available knowledge on our subject of interest, we conducted the broad database search last to ensure our search results were comprehensive.

- (pg.6) PROSPERO is a systematic review protocol registry, does the team mean PRISMA?

We indeed meant PRISMA-S. We replaced PROSPERO with PRISMA in the manuscript. Thank you for finding this error (Line 149).

- please review PRISMA-S and report search strategies accordingly. They should be provided in full as run in the supplemental files. For example from how the searches are presented, it’s hard to know if subject headings were searched in addition to keywords, what fields were searched, etc.

We now include the PRISMA-S checklist and all search strategies in the supplementary material (see Supplementary Material 1).

- I'm a little concerned that the search may be overly simplistic (see for example the racial/ethnic search filters available here: https://sites.google.com/a/york.ac.uk/issg-search-filters-resource/home/population-specific).

We agree that this is a valid concern given the complexity of defining and capturing racial and ethnic disparities in healthcare research. However, our approach was guided by our specific research objective: to identify interventions addressing racial and ethnic disparities in critical care, rather than disparities affecting a particular group. Because of this broad scope, we designed a search strategy that was inclusive of all relevant studies, regardless of the population examined. To ensure methodological rigor, we collaborated with a research librarian, Hope Lappen, MLIS, MS, who has expertise in systematic and scoping review search strategies. Her input refined our search strategy, ensuring that it captured the breadth of relevant literature without being overly narrow or missing key studies.

Results:

- I really like the way the team chose to break down results. It's very clear and the groupings are helpful for getting a bird's eye view.

- your tables are really clear and easy to read

Thank you for these kind comments on our Results and Tables.

Discussion:

- the Kilbourne phases is a helpful breakdown

- I think it's important here to emphasize that these are only American findings, and there may be different levels of development/interrogation of mechanism of action in other jurisdictions that are similar to the US (e.g. Canada, UK, Australia), or just globally, since every country has their own racial minorities. It would be helpful to contextualize your findings with the rest of the world, or if that's not feasible, frequently emphasize that these findings only relate to the US and not all of critical care literature.

We now emphasize that these findings are specific to the US and discuss the rationale for and limitations inherent in this approach (Lines 369-375).

- quality appraisal in scoping reviews are not common because they're intended to answer different questions. It might be simpler to just cite JBI here rather than getting into breadth of study designs.

Thank you for this suggestion; we now cite JBI to support this point (Line 302-303, Reference #28).

- I think your decision to limit to US-only is also a limitation, since NICUs/ICUs are pretty structured settings (it's not like we're comparing public health initiatives or something really nebulous) and could have had a lot to offer and provided richer context of where the literature is right now.

We now address the US-only inclusion criterion as a limitation in the Discussion.

Final thoughts: This was a really well written and thoughtful paper, and provides some very rich feedback for researchers in this field moving forward. I'm a bit concerned about the robustness of the search, especially if subject headings weren't used, as that is standard practice for knowledge synthesis searching. This would be easy enough to fix, though it would take this from a minor revisions to a major one since it would require additional screening, etc.

We would like to thank the reviewer for their insights and suggestions. The subject headings of literature were included in the search; this is now evident in the supplementary material detailing all searches (Supplementary Material 2) and it is now mentioned in the Methods section (Lines 106-107).

---

## [Decision Letter · Decision Letter 1]

12 Apr 2025

Dear Dr. Ge,

Thank you for submitting your manuscript to PLOS ONE. After careful consideration, we feel that it has merit but does not fully meet PLOS ONE’s publication criteria as it currently stands. Therefore, we invite you to submit a revised version of the manuscript that addresses the points raised during the review process.

We look forward to receiving your revised manuscript.

Kind regards,

De-Chih Lee, Ph.D.

Academic Editor

PLOS ONE

Additional Editor Comments :

Please make major revisions based on the review by two reviewers. Authors are asked to review the questions raised by the first reviewer to find if any critical literature has been missed.

Reviewers' comments:

Reviewer's Responses to Questions

**Comments to the Author**

Reviewer #2: (No Response)

Reviewer #3: (No Response)

2. Is the manuscript technically sound, and do the data support the conclusions?

Reviewer #2: Yes

Reviewer #3: Yes

3. Has the statistical analysis been performed appropriately and rigorously?

Reviewer #2: N/A

Reviewer #3: Yes

4. Have the authors made all data underlying the findings in their manuscript fully available?

Reviewer #2: No

Reviewer #3: Yes

5. Is the manuscript presented in an intelligible fashion and written in standard English?

Reviewer #2: Yes

Reviewer #3: Yes

Reviewer #2: Thank you very much for all the work you have put into revising this paper, already it is stronger!

Unfortunately, now that I can see the searches in full, they are not sufficiently robust to be published as part of a scoping review. The Web of Science search is fine as the database doesn't have subject headings, but PubMed, Embase and CINAHL all have subject headings that would directly apply to this topic (e.g. in PubMed: Ethnicity, Minority Groups, Race Factors, Racial Groups), and the use of both keywords and subject headings is recommended by all major synthesis bodies (Cochrane, JBI, etc). Some of these databases have mapping capabilites but they are not robust - when I tested the racial component of the search in PubMed and CINAHL the results were different with subject headings included. Here is an example of a broad race/ethnicity search for context (though it is for Ovid Medline): https://hsls.libguides.com/Ovid-Medline-search-filters/race-ethnicity#s-lg-box-32796073. The same goes for disparities, ICU, and to a lesser extent, quality improvement. All aspects of the search should have subject headings as well as keywords where applicable subject headings exist.

In addition to the fact that the search does not make use of subject headings, there are still numerous terms that I would expect to see in each category: minority group*, people/person/communit*/children/men/women of color/colour, equity, unequit*, unequal*, inadequat*, underserv*, disadvant*, depriv*, disproport*, bias, discriminat*, prejud*, racism, racist, potentially barrier*, critically ill, ECMO and ARDS acronyms spelled out, septic, potentially facilitat*

I recommend the team revisit the searches with the librarian with an eye for subject headings and keywords (which you could limit to title/abstract only to keep numbers reasonable, as this is common practice for reviews), overlap this with what you've already screened and screen the difference.

Reviewer #3: Very important and necessary contribution. I appreciated reading the in-depth description of how the team developed the taxonomy as well as how you decided to include or exclude studies (i.e. studies that had the 'wrong intervention').

Please reframe the next steps (i.e. instead of calling for randomized control trials since it is unethical to perform this type of research in the social sciences by exposing a group to poor conditions for the purpose of investigation). Also, scoping reviews are not designed to assess the quality of research but the breadth so please update the statement that you conducted a rigorous evaluation of interventions aimed at eliminating disparities in critical care.

The abstract notes there were 86 studies that underwent full text review but the narrative lists that number as 75. Which was it?

Since you cite the AMA's guidance on the need for the reviewed studies to define and frame race, will you elucidate the reason that your research team did not decide to also follow the AMA guidance on the reporting of race in your narrative (i.e. lowercase the 'w' in white): https://www.ama-assn.org/system/files/ama-aamc-equity-guide.pdf

Otherwise, well done.

**Do you want your identity to be public for this peer review?** For information about this choice, including consent withdrawal, please see our Privacy Policy

Reviewer #2: No

Reviewer #3: No

---

## [Author Response · Author response to Decision Letter 2]

3 Jun 2025

Emily Chenette, PhD

Editor-in-Chief

PLOS One

June 2, 2025

Dear Dr. Chenette and the PLOS One Editorial Team,

Thank you for your consideration of our revised manuscript “Interventions to improve racial and ethnic equity in critical care: A scoping review”. We are grateful for the reviewers’ thoughtful and constructive feedback. We have considered each of their comments and have made the necessary revisions. Below, we provide a point-by-point response to the reviewers’ concerns. We believe these revisions have improved the rigor and impact of our manuscript. We hope that our revised manuscript now meets the standards for publication in PLOS One and look forward to your assessment.

Reviewer #2: Thank you very much for all the work you have put into revising this paper, already it is stronger!

Comment: Unfortunately, now that I can see the searches in full, they are not sufficiently robust to be published as part of a scoping review. The Web of Science search is fine as the database doesn't have subject headings, but PubMed, Embase and CINAHL all have subject headings that would directly apply to this topic (e.g. in PubMed: Ethnicity, Minority Groups, Race Factors, Racial Groups), and the use of both keywords and subject headings is recommended by all major synthesis bodies (Cochrane, JBI, etc). Some of these databases have mapping capabilites but they are not robust - when I tested the racial component of the search in PubMed and CINAHL the results were different with subject headings included. Here is an example of a broad race/ethnicity search for context (though it is for Ovid Medline): https://hsls.libguides.com/Ovid-Medline-search-filters/race-ethnicity#s-lg-box-32796073. The same goes for disparities, ICU, and to a lesser extent, quality improvement. All aspects of the search should have subject headings as well as keywords where applicable subject headings exist.

In addition to the fact that the search does not make use of subject headings, there are still numerous terms that I would expect to see in each category: minority group*, people/person/communit*/children/men/women of color/colour, equity, unequit*, unequal*, inadequat*, underserv*, disadvant*, depriv*, disproport*, bias, discriminat*, prejud*, racism, racist, potentially barrier*, critically ill, ECMO and ARDS acronyms spelled out, septic, potentially facilitat*

I recommend the team revisit the searches with the librarian with an eye for subject headings and keywords (which you could limit to title/abstract only to keep numbers reasonable, as this is common practice for reviews), overlap this with what you've already screened and screen the difference.

Response: Co-author research librarian Hope Lappen revised and re-ran the search strategy to incorporate subject headings and additional keywords as suggested by the Reviewer. PubMed, Embase, and CINAHL were queried again with the new search strategy that included subject headings and the new keywords. Web of Science was also re-queried with the new search strategy to include the additional keywords. The Methods section is updated to reflect the new strategy (lines 101-128) and the revised search strategy for each database can be found in the updated Supplementary Material 2. An additional 813 studies were imported for screening, 39 duplicates were removed, resulting in an additional 774 studies screened. No additional studies meeting inclusion criteria were found (line 165). The PRISMA Flowchart (Figure 1) and the list of studies excluded in full-text review (S3) were updated accordingly.

Reviewer #3: Very important and necessary contribution. I appreciated reading the in-depth description of how the team developed the taxonomy as well as how you decided to include or exclude studies (i.e. studies that had the 'wrong intervention').

Response: Thank you for this kind feedback on our study analysis.

Comment: Please reframe the next steps (i.e. instead of calling for randomized control trials since it is unethical to perform this type of research in the social sciences by exposing a group to poor conditions for the purpose of investigation). Also, scoping reviews are not designed to assess the quality of research but the breadth so please update the statement that you conducted a rigorous evaluation of interventions aimed at eliminating disparities in critical care.

Response: For next steps, we suggested increased randomized control trials (RCTs, lines 314-322) because RCTs are generally the gold-standard study type to assess if an intervention is effective. Similar to the RCTs that met inclusion criteria in the scoping review, RCTs would expose study groups to a novel intervention aimed at reducing a pre-existing disparity in patient outcomes, instead of exposing a group to poor conditions to simulate a disparity. To illustrate, the RCT “Reducing Disparity in Receipt of Mother's Own Milk (ReDiMOM)” (Reference 34) randomized mother-infant pairs to either standard lactation care or an intervention designed to increase use of mother’s own milk and reduce disparities in breastfeeding rates between Black and non-Black infants. We highlight designs such as stepped-wedge cluster-randomized trials that allow all participating ICUs to benefit from the intervention under study (lines 318-320).

For assessing quality of research, we agree with the Reviewer as we discussed how we “did not seek to assess the quality of study designs because we aimed to survey current knowledge on reducing disparities in the ICU and, as our review demonstrates, research to reduce disparities is at an early state” (lines 317-318). In the Abstract, we did not aim to say that the scoping review was a rigorous evaluation of interventions, but rather that we only found a few interventions have been employed to address disparities in critical care. The Abstract has been edited to clarify this point (line 57).

Comment: The abstract notes there were 86 studies that underwent full text review but the narrative lists that number as 75. Which was it?

Response: The narrative is consistent with the abstract as eighty-six studies underwent full-text review from which 75 studies were excluded (lines 50, 188, and 193 in the tracked changes manuscript). In the revised search in response to Reviewer #2, 93 studies underwent full-text review from which 82 studies were excluded (lines 49, 165, and 168).

Comment: Since you cite the AMA's guidance on the need for the reviewed studies to define and frame race, will you elucidate the reason that your research team did not decide to also follow the AMA guidance on the reporting of race in your narrative (i.e. lowercase the 'w' in white): https://www.ama-assn.org/system/files/ama-aamc-equity-guide.pdf

Otherwise, well done.

Response: The AMA-AAMC Equity Guide that is linked in the Reviewer’s comment is a different citation than the one cited in the manuscript, Updated Guidance on the Reporting of Race and Ethnicity in Medical and Science Journals. JAMA. 2021;326(7):621-7 (Reference 52). The Equity Guide to which the Reviewer refers is a communication handbook for clinical practice for when physicians are speaking and referring to patients and documenting care. Our cited reference informs the updated guidelines in reporting race and ethnicity for medical and scientific publications, as found in the eleventh edition of AMA Manual of Style: A Guide for Authors and Editors. We thus followed the AMA Manual of Style and capitalized the names of races including White.

We again sincerely appreciate the time and effort of the reviewers in providing thoughtful feedback on our manuscript. We hope that our revisions address the reviewers’ concerns and that our manuscript is now suitable for publication in PLOS One. Thank you for your consideration and we look forward to your response.

Sincerely,

Mari Armstrong-Hough, PhD, MPH

Associate Professor of Epidemiology and Social & Behavioral Sciences

New York University

+1 (212) 998-9015 | mah842@nyu.edu

Shirley Ge, MD

Research Associate, AIRE Lab

New York University

+1 (978) 407-1314 | shirleyge255@gmail.com

---

## [Decision Letter · Decision Letter 2]

16 Jul 2025

Dear Dr. Ge,

Thank you for submitting your manuscript to PLOS ONE. After careful consideration, we feel that it has merit but does not fully meet PLOS ONE’s publication criteria as it currently stands. Therefore, we invite you to submit a revised version of the manuscript that addresses the points raised during the review process.

The authors are requested to accept the paper after minor revisions based on the reviewers' comments.

We look forward to receiving your revised manuscript.

Kind regards,

De-Chih Lee, Ph.D.

Academic Editor

PLOS ONE

Journal Requirements:

Additional Editor Comments (if provided):

The authors are requested to accept the paper after minor revisions based on the reviewers' comments.

Reviewers' comments:

Reviewer's Responses to Questions

**Comments to the Author**

Reviewer #3: All comments have been addressed

Reviewer #4: (No Response)

2. Is the manuscript technically sound, and do the data support the conclusions?

Reviewer #3: Yes

Reviewer #4: Yes

3. Has the statistical analysis been performed appropriately and rigorously?

Reviewer #3: Yes

Reviewer #4: N/A

4. Have the authors made all data underlying the findings in their manuscript fully available?

Reviewer #3: Yes

Reviewer #4: Yes

5. Is the manuscript presented in an intelligible fashion and written in standard English?

Reviewer #3: Yes

Reviewer #4: Yes

Reviewer #3: Great job with responding to the reviewer comments and sufficiently editing the manuscript .

Reviewer #4: Dear Authors

Great work in submitting the revised version of the manuscript. A few comments to address,

The authors have suggested the need to randomized, controlled evaluations of interventions to promote equity in critical care – However, to design a well-controlled and adequately powered RCT the cost and resources, along with the sample size will be enormous. How about considering a SWAT (Studies within a trial) to identify racial and ethic differences in interventions (just a suggestion)

Line 67, references 7 to 9 – quotes one of the author’s earlier papers, I am wondering if this will that be a potential conflict of interest for this scoping review?

Line 84, 85 – “to consolidate the available knowledge regarding interventions and quality improvement initiatives to reduce racial and ethnic disparities in critical care.”

I am not sure how much of the second objective has been satisfied with this review, it might be helpful to highlight this one.

The authors have attempted to address the racial and ethnic disparities in the studies – it might be helpful to add how much was the ethnic variability in the study team, might be interesting from a reader perspective to know about the team (please ignore this if you think it’s inappropriate)

I am not sure if PRISMA standard or PRISMA ScR checklist is suitable for use in scoping reviews?

The authors have aimed to addressed the disparities in US – just curious to know why, a few statements to strengthen why studying about US might be helpful.

In the results section – it might be helpful to mention the race and ethnicity of the population studied in the included studies.

It might be helpful to highlight the ongoing studies in table 1 with an easy identifier (different color or *) for easy identification.

Table 1 – it might be helpful to add the total number of sample size along with the % of race and ethnicity within the studies.

Table 2 is something I really liked. Well done.

It might be better to add the strengths of the study before the limitations.

**Do you want your identity to be public for this peer review?** For information about this choice, including consent withdrawal, please see our Privacy Policy

Reviewer #3: No

Reviewer #4: No

---

## [Author Response · Author response to Decision Letter 3]

29 Aug 2025

Emily Chenette, PhD

Editor-in-Chief

PLOS One

August 11, 2025

Dear Dr. Chenette and the PLOS One Editorial Team,

Thank you for your consideration of our revised manuscript “Interventions to improve racial and ethnic equity in critical care: A scoping review”. We are grateful for the reviewers’ thoughtful and constructive feedback. We have considered each of their comments and have made the necessary revisions. Below, we provide a point-by-point response to the reviewers’ concerns. We believe these revisions have improved the rigor and impact of our manuscript. We hope that our revised manuscript now meets the standards for publication in PLOS One and look forward to your assessment.

Reviewer #3: Great job with responding to the reviewer comments and sufficiently editing the manuscript .

Thank you for this positive feedback.

Reviewer#4: Dear Authors

Great work in submitting the revised version of the manuscript. A few comments to address,

The authors have suggested the need to randomized, controlled evaluations of interventions to promote equity in critical care – However, to design a well-controlled and adequately powered RCT the cost and resources, along with the sample size will be enormous. How about considering a SWAT (Studies within a trial) to identify racial and ethic differences in interventions (just a suggestion)

We appreciate the reviewer’s suggestion to consider Studies Within a Trial (SWAT) as an alternative to large randomized controlled trials. While SWAT’s can identify subgroup differences within ongoing trials, we discussed how our results show that there is a paucity of intervention studies aiming to reduce racial or ethnic disparities in critical care despite there are already being many studies that detect racial or ethnic disparities in the ICU (Lines 269-291). Moreover, many of the studies meeting inclusion criteria focused on increasing advantageous health behaviors for all patients in practices that have been shown to have race or ethnic-based differences. These studies aimed to promote healthy behavior and then measured the impact on patients stratified by race, similar to the reviewer’s suggestion of SWAT’s (Lines 203-209).

Line 67, references 7 to 9 – quotes one of the author’s earlier papers, I am wondering if this will that be a potential conflict of interest for this scoping review?

We do not feel citing a co-author’s previous paper, Ethnic Disparities in Deep Sedation of Patients with Acute Respiratory Distress Syndrome in the United States: Secondary Analysis of a Multicenter Randomized Trial (reference 9), is a conflict of interest because we cite the paper to provide background on the evidence for ethnic disparities in ICU care in addition to several other examples from other papers by different authors (Lines 64-70, references 1-15). This paper is not included in the scoping review itself, nor would it be eligible because it does not report or evaluate an intervention.

Line 84, 85 – “to consolidate the available knowledge regarding interventions and quality improvement initiatives to reduce racial and ethnic disparities in critical care.”

I am not sure how much of the second objective has been satisfied with this review, it might be helpful to highlight this one.

We only had one primary objective to consolidate the current knowledge on interventions and quality improvement initiatives to reduce racial ethnic disparities in critical care by reviewing existing and ongoing studies (Lines 82-85). Table 1 summarizes our findings of the current knowledge and we comment on the implications of our results in the Discussion (Lines 268-351).

The authors have attempted to address the racial and ethnic disparities in the studies – it might be helpful to add how much was the ethnic variability in the study team, might be interesting from a reader perspective to know about the team (please ignore this if you think it’s inappropriate)

Our objective was to synthesize evidence on interventions to address racial and ethnic disparities in critical care, and we believe the focus should remain on the content and quality of the included studies rather than on the demographics of the review team.

I am not sure if PRISMA standard or PRISMA ScR checklist is suitable for use in scoping reviews?

Thank you for this feedback. We have replaced the PRIMSA standard checklist with the PRISMA extension for scoping reviews (PRISMA-ScR) as PRISMA-ScR is specifically for scoping reviews (Supplementary Material 1).

The authors have aimed to addressed the disparities in US – just curious to know why, a few statements to strengthen why studying about US might be helpful.

Non-U.S. studies were excluded because racial and ethnic disparities in healthcare in the United States arise from a distinct combination of historical, policy, and sociocultural factors. Studies conducted in other countries reflect disparities shaped by their own sociopolitical environments and healthcare systems, which limits the applicability and comparability of their findings to the U.S. context (Lines 361-365).

In the results section – it might be helpful to mention the race and ethnicity of the population studied in the included studies.

It might be helpful to highlight the ongoing studies in table 1 with an easy identifier (different color or *) for easy identification.

Table 1 – it might be helpful to add the total number of sample size along with the % of race and ethnicity within the studies.

Table 2 is something I really liked. Well done.

It might be better to add the strengths of the study before the limitations.

Thank you to the reviewer for these suggestions. To table 1, we added indicators for ongoing clinical trials and studies with interventions that continued to be implemented past the study period with “*” and “†”, respectively. Table 1 already includes a column for study population. Regarding the sample size and proportion of race and ethnicity within each study population, the specific demographics of participants, beyond the targeted population already included in the table, are not relevant to answering the study objective. We sought to consolidate the current knowledge on interventions and quality improvement initiatives to reduce racial ethnic disparities in critical care by reviewing existing and ongoing studies in a scoping review (Lines 82-85). Unlike a systematic review, demographics and outcomes of included studies are not always necessary to answer the research question.

We again sincerely appreciate the time and effort of the reviewers in providing thoughtful feedback on our manuscript. We hope that our revisions address the reviewers’ concerns and that our manuscript is now suitable for publication in PLOS One. Thank you for your consideration and we look forward to your response.

Sincerely,

Mari Armstrong-Hough, PhD, MPH

Associate Professor of Epidemiology and Social & Behavioral Sciences

New York University

+1 (212) 998-9015 | mah842@nyu.edu

Shirley Ge, MD

Research Associate, AIRE Lab

New York University

+1 (978) 407-1314 | shirleyge255@gmail.com

---

## [Decision Letter · Decision Letter 3]

3 Oct 2025

Dear Dr. Ge,

Thank you for submitting your manuscript to PLOS ONE. After careful consideration, we feel that it has merit but does not fully meet PLOS ONE’s publication criteria as it currently stands. Therefore, we invite you to submit a revised version of the manuscript that addresses the points raised during the review process.

Please make minor revisions according to the second reviewer's comments and then accept the revised manuscript.

We look forward to receiving your revised manuscript.

Kind regards,

De-Chih Lee, Ph.D.

Academic Editor

PLOS ONE

Journal Requirements:

Additional Editor Comments:

Please make minor revisions according to the second reviewer's comments and then accept the revised manuscript.

Reviewers' comments:

Reviewer's Responses to Questions

**Comments to the Author**

Reviewer #3: All comments have been addressed

Reviewer #4: All comments have been addressed

2. Is the manuscript technically sound, and do the data support the conclusions?

Reviewer #3: Yes

Reviewer #4: Yes

3. Has the statistical analysis been performed appropriately and rigorously?

Reviewer #3: No

Reviewer #4: N/A

4. Have the authors made all data underlying the findings in their manuscript fully available?

Reviewer #3: Yes

Reviewer #4: Yes

5. Is the manuscript presented in an intelligible fashion and written in standard English?

Reviewer #3: Yes

Reviewer #4: Yes

Reviewer #3: Great job responding in detail to all of the reviewer comments thus far. One last question I have is why your team did not elect to rely on 3 researchers for the screening/review of studies (2 initial reviewers and 1 to break ties). This is the standard way to pursue interrater reliability. Please add a line as to why you opted to rely on 2 researchers rather than 3.

Reviewer #4: Dear Authors,

The submitted manuscript reads very well.

Please modify the PRISMA to PRISMA ScR (more than two places still read as PRISMA). In the discussion, I will encourage you to add the strengths of the study before the limitations (content from line 366-371 may be placed before line 352).

All the best!

**Do you want your identity to be public for this peer review?** For information about this choice, including consent withdrawal, please see our Privacy Policy

Reviewer #3: **Yes: ** Maranda Ward

Reviewer #4: No

---

## [Author Response · Author response to Decision Letter 4]

20 Oct 2025

Dear Dr. Chenette and the PLOS One Editorial Team,

Thank you for your consideration of our revised manuscript “Interventions to improve racial and ethnic equity in critical care: A scoping review”. We are grateful for the reviewers’ thoughtful and constructive feedback. We have considered each of their comments and have made the necessary revisions. Below, we provide a point-by-point response to the reviewers’ concerns. We believe these revisions have improved the rigor and impact of our manuscript. We hope that our revised manuscript now meets the standards for publication in PLOS One and look forward to your assessment.

Reviewer #3: Great job responding in detail to all of the reviewer comments thus far. One last question I have is why your team did not elect to rely on 3 researchers for the screening/review of studies (2 initial reviewers and 1 to break ties). This is the standard way to pursue interrater reliability. Please add a line as to why you opted to rely on 2 researchers rather than 3.

Research librarian Hope Lappen was often consulted as the third reviewer on studies for which the initial two reviewers had difficulty reaching a consensus. A consensus was reached with Lappen’s expertise or with Lappen casting a tie-breaking vote. Her role as the third reviewer was added to the Methods section (Line 139-140).

Reviewer #4: Dear Authors,

The submitted manuscript reads very well.

Please modify the PRISMA to PRISMA ScR (more than two places still read as PRISMA). In the discussion, I will encourage you to add the strengths of the study before the limitations (content from line 366-371 may be placed before line 352).

All the best!

PRISMA was modified to PRISMA-ScR in the manuscript text and in the title of Figure 1. The discussion of the study strengths was moved before the limitations.

We again sincerely appreciate the time and effort of the reviewers in providing thoughtful feedback on our manuscript. We hope that our revisions address the reviewers’ concerns and that our manuscript is now suitable for publication in PLOS One. Thank you for your consideration and we look forward to your response.

Sincerely,

Mari Armstrong-Hough, PhD, MPH

Associate Professor of Epidemiology and Social & Behavioral Sciences

New York University

+1 (212) 998-9015 | mah842@nyu.edu

Shirley Ge, MD

Research Associate, AIRE Lab

New York University

+1 (978) 407-1314 | shirleyge255@gmail.com

---

## [Decision Letter · Decision Letter 4]

2 Nov 2025

Interventions to improve racial and ethnic equity in critical care: A scoping review

PONE-D-24-49820R4

Dear Dr. Ge,

We’re pleased to inform you that your manuscript has been judged scientifically suitable for publication and will be formally accepted for publication once it meets all outstanding technical requirements.

Kind regards,

De-Chih Lee, Ph.D.

Academic Editor

PLOS ONE

Additional Editor Comments (optional):

Reviewers' comments:

Reviewer's Responses to Questions

**Comments to the Author**

Reviewer #3: All comments have been addressed

Reviewer #4: All comments have been addressed

2. Is the manuscript technically sound, and do the data support the conclusions?

Reviewer #3: Yes

Reviewer #4: Yes

3. Has the statistical analysis been performed appropriately and rigorously?

Reviewer #3: Yes

Reviewer #4: N/A

4. Have the authors made all data underlying the findings in their manuscript fully available?

Reviewer #3: Yes

Reviewer #4: Yes

5. Is the manuscript presented in an intelligible fashion and written in standard English?

Reviewer #3: Yes

Reviewer #4: Yes

Reviewer #3: Authors thoroughly and carefully responded to all of the reviewer comments. Very well done, look forward to this publication being widely available.

Reviewer #4: Dear Authors

Great job in making the changes. I would suggest you to change reference 27 in the article to this one please "Tricco AC, Lillie E, Zarin W, O'Brien KK, Colquhoun H, Levac D, Moher D, Peters MDJ, Horsley T, Weeks L, Hempel S, Akl EA, Chang C, McGowan J, Stewart L, Hartling L, Aldcroft A, Wilson MG, Garritty C, Lewin S, Godfrey CM, Macdonald MT, Langlois EV, Soares-Weiser K, Moriarty J, Clifford T, Tunçalp Ö, Straus SE. PRISMA Extension for Scoping Reviews (PRISMA-ScR): Checklist and Explanation. Ann Intern Med. 2018 Oct 2;169(7):467-473. doi: 10.7326/M18-0850. Epub 2018 Sep 4. PMID: 30178033."

All the best!

**Do you want your identity to be public for this peer review?** For information about this choice, including consent withdrawal, please see our Privacy Policy

Reviewer #3: **Yes: ** Maranda C. Ward

Reviewer #4: No

---

## [Editor Report · Acceptance letter]

PONE-D-24-49820R4

PLOS ONE

Dear Dr. Ge,

I'm pleased to inform you that your manuscript has been deemed suitable for publication in PLOS ONE. Congratulations! Your manuscript is now being handed over to our production team.

Kind regards,

on behalf of

Dr. De-Chih Lee

Academic Editor

PLOS ONE